# A New Approach to Imaging and Rapid Microbiome Identification for Prostate Cancer Patients Undergoing Radiotherapy

**DOI:** 10.3390/biomedicines10081806

**Published:** 2022-07-27

**Authors:** Ewelina Maślak, Wioletta Miśta, Michał Złoch, Dominika Błońska, Paweł Pomastowski, Fernanda Monedeiro, Bogusław Buszewski, Jolanta Mrochem-Kwarciak, Katarzyna Bojarska, Dorota Gabryś

**Affiliations:** 1Centre for Modern Interdisciplinary Technologies, Nicolaus Copernicus University, Wileńska 4 Str., 87-100 Toruń, Poland; e_maslak@doktorant.umk.pl (E.M.); michal.zloch@umk.pl (M.Z.); dominikab10@gmail.com (D.B.); pomastowski.pawel@gmail.com (P.P.); fmonedeiro@gmail.com (F.M.); bbusz@chem.umk.pl (B.B.); 2Department of Environmental Chemistry and Bioanalytics, Faculty of Chemistry, Nicolaus Copernicus University, Gagarina 7 Str., 87-100 Toruń, Poland; 3Radiotherapy Department, Maria Sklodowska-Curie National Research Institute of Oncology, Gliwice Branch, Wybrzeże Armii Krajowej 15 Str., 44-102 Gliwice, Poland; wioletta.mista@io.gliwice.pl; 4Analytics and Clinical Biochemistry Department, Maria Sklodowska-Curie National Research Institute of Oncology, Gliwice Branch, Wybrzeże Armii Krajowej 15 Str., 44-102 Gliwice, Poland; jolanta.mrochem-kwarciak@io.gliwice.pl (J.M.-K.); katarzyna.bojarska@io.gliwice.pl (K.B.)

**Keywords:** microbiome, prostate cancer, radiotherapy, MALDI

## Abstract

(1) Background: Little is known about the impact of urinary microflora, in particular, its effects on side effects after radiotherapy. The use of mass spectrometry identification method (MALDI) may bring a new look at the issue of the composition and significance of the urinary microbiome. This study aimed to use the mass spectrometry identification method (MALDI) to identify the microbiome of urine samples collected from 50 irradiated prostate cancer patients. (2) Methods: Blood and urine samples were collected before gold marker implantation, at the start and last day of radiotherapy, 1, 4 months after. Patients do not always collect the urine from the midstream; therefore, samples were collected from the first void and midstream in 12 patients for MALDI analysis; in the remaining 38 patients—from the midstream void for MALDI and biochemical analysis. (3) Results: Microorganisms were present in 140/181 urine samples. We found 33 different species 3G(−) and 30G(+). The most frequently isolated strains were: *Staphylococcus haemolyticus*, *Staphylococcus epidermidis*, *Staphylococcus hominis*, *Enterococcus faecalis*, and *Micrococcus luteus*. When comparing the type of urine samples, bacteria were more common in samples from the first-void urine than from the midstream one. The absence of bacteria was found in 12.2% of samples from the first-void urine and in 24.7% from the midstream. There was no difference in the total incidence of species between streams (*p* = 0.85). Before fiducial implantation, the total number of detected bacterial species was significantly higher in comparison to the end of radiotherapy (*p* = 0.038), indicating that the administered therapy resulted in depleting the local microbiome. One month after radiotherapy, an increase in the number of isolated bacteria was observed. The number of bacterial species in urine did not correlate with blood parameters. The presence of leukocytes (*p* = 0.013) and proteins (*p* = 0.004) in urine was related to a greater variety of bacteria found in urine specimens. (4) Conclusions: We obtained a similar spectrum of bacteria from the initial and middle urine streams. We also showed that there is a change in bacteria species affected by the treatment of prostate cancer patients, with both antibiotics before gold fiducial implantation and radiotherapy.

## 1. Introduction

Conducting radiotherapy in the pelvic area is accompanied by the occurrence of early and late complications. The type and severity of radiotherapy side effects depend on the dose, on the amount of healthy tissue that is exposed to the radiation, and they are patient specific. Newer technologies, such as intensity-modulated radiation therapy (IMRT), deliver the highest dose of radiation to the target while sparing the surrounding healthy tissue. This helps minimize the side effects of external beam radiation treatment. Still, some of the urinary tract side effects may significantly decrease patient quality of life, such as frequent, difficult, or painful urination, urinary leakage, blood in urine, abdominal cramping, nocturia.

Radiobiological mechanisms of complications are known concerning cell killing by irradiation and tissue response to treatment [1]. Unfortunately, little is known about the effects of urinary microflora on these types of side effects present after radiation treatment or their role in clinical outcomes. The microbiome, beyond its role in tumorigenesis and cancer progression, can be used as a new potential biomarker in the diagnosis, prognosis, and risk stratification of this disease [2]. Currently, there are many kinds of research focused on searching for microbiota biomarkers in cancer for many urogenital and systemic pathologies [3]. Results from numerous studies suggest that there is a link between cancer, microbiota, and their influence on response to therapeutic efficacy. There are two approaches to studying the relationship between the microbiome and prostate cancer: direct (prostate tissue or urine analysis) and indirect (fecal analysis) [4]. However, most of these papers are focused on gut microbiota because, until recently, any growth of bacteria from urine was considered invasive, and later, it was considered unethical to obtain urine from a bladder biopsy or suprapubic aspiration from healthy individuals to characterize its microbiome while avoiding contamination of the sample with urethral microorganisms [5]. In addition, many bacterial species do not show growth using standard techniques. Fortunately, the advances in molecular biology techniques and culture methods have enabled the definition of a urinary microbiome associated with urine most commonly obtained from the midstream [6,7,8].

Nevertheless, there is still a lack of studies describing the importance of the urinary microbiome in radiation-induced changes in the present literature. The majority of the studies have been conducted to provide a relation between the urinary microbiome and urinary tract infections. In most studies, the identification of the microbiome was performed using the 16S rRNA gene sequencing technique, which is the one most commonly used to identify microorganisms in urine samples [9,10]. Additionally, expanded quantitative urine culture (EQUC) is used for urine microbiome identification, which uses a variety of urine volumes, culture media, and incubation conditions to grow bacteria, and quantitative polymerase chain reaction (PCR), which provides quantified information about microorganisms [11]. Proteomics also offers methods for microbial identification. More recently, matrix-assisted laser desorption ionization time of flight mass spectrometry (MALDI TOF MS) has been proven to be a rapid and reliable tool for identifying a wide array of microbial species, with accuracy comparable to that of PCR analysis [12,13]. Therefore, this technique has emerged as a key tool in pathogen identification methods used in clinical laboratories in recent years.

The present study aimed to use the MALDI technique to identify the microbiome and evaluate its changes in urine samples collected from patients treated for prostate cancer undergoing radiotherapy. The compositions of the microbiome were analyzed at various time points before fiducial implantation to see the original bacterial compositions in the urine (before antibiotics were used); then, at the beginning and end of radiotherapy; and during the follow-up, 1 month and 4 months after the end of irradiation. In addition, changes in the species composition of microorganisms at different stages of radiotherapy and the correlation of the obtained results with alteration in immune mediators (i.e., lymphocytes) present in blood and urine samples were investigated.

## 2. Materials and Methods

### 2.1. Patient Characteristics

Patients qualified primarily or post-operatively for radical irradiation for prostate cancer at all stages, with or without hormonal treatment, with or without gold fiducial implantation. One to three gold fiducials are small gold coils, about the size of a grain of rice, which are placed into the prostate gland of patients treated primarily for prostate cancer.

Prostate cancer radical radiation therapy and radiation treatment planning were conducted at the Maria Sklodowska-Curie National Research Institute of Oncology in accordance with the valid protocol. Patient characteristics are shown in Table 1.

The treatment decision was made according to disease stage, histopathological grade, and PSA level. Participation in the study did not affect the choice of method, irradiated volumes, and dose. Thirty patients were on hormonal treatment as analog LH-RH, with or without Flutamide. Radiotherapy was delivered on a linear accelerator (25 pts) or cyberknife (25 pts). The volumes included: prostate alone, prostate with a base of seminal vesicles, prostate with the seminal vesicles (primary treatment), irradiated to a total dose of 76–78 Gy delivered in 2 Gy/fraction standard fractionation, or hypofractionation of 36.25 Gy delivered in 7.25 Gy/fraction, using MV photons; prostate bed (patients after radical prostatectomy), irradiated to a total dose of 66–76 Gy delivered in 2 Gy/fraction using MV photons. One patient had a prostate boost of 15 Gy delivered with brachytherapy. If required, pelvic lymph nodes were irradiated to a total dose of 44–50 Gy delivered in 2 Gy/fraction using MV photons, with or without a boost to the involved lymph nodes to a total dose of 60–68 Gy, or delivered as a stereotactic boost of 16 Gy in 2 fractions. Appropriate regulatory approval, including ethical approval, was obtained in all jurisdictions. The study was approved by the NIO-PIB Ethics Committee KB/430-104/19 and conducted in accordance with the principles of Good Clinical Practice. All patients provided written informed consent.

### 2.2. Urine and Blood Samples

Urine and blood samples were prospectively collected from patients; a total of 181 urine samples were obtained from 50 patients undergoing radiotherapy of the pelvic area at different time points: t1—before gold fiducial implantation into the prostate gland (43), t2—before radiotherapy (49), t3—at the end of radiotherapy (42), at follow-up; t4—1 month after radiotherapy (36), t5—4 months after radiotherapy (11). At time point t1, patients had blood and urine collected at least 1 day before starting antibiotics and prior rectal debridement. Usually, Azithromycin 500 mg or Ciprofloxacin 250 mg was prescribed. An enema infusion the evening before and the morning of the day of gold marker implantation was usually recommended for cleaning the rectum before the procedure.

It is recommended to collect urine from the midstream, but patients do not always do this correctly despite being instructed. Information for the patient midstream collection for microbiological studies is shown in Appendix A. Therefore, we decided to collect samples from the first void and midstream in the first 12 patients for MALDI analysis; both samples were frozen at −80 °C and later sent to the Centre for Modern Interdisciplinary Technologies in Toruń. From the remaining 38 patients, we collected two samples from the midstream-void one for MALDI and the second one for biochemical analysis. After sampling, one container with urine was handed over to the diagnostic laboratory for routine tests, and the second one was frozen at −80 °C and later sent to the Centre for Modern Interdisciplinary Technologies in Toruń.

### 2.3. Biochemical Tests of Blood and Urine

For urine samples, general tests were performed, such as color, transparency, pH, specific gravity, and standard urine culture, performed obligatorily in each patient during radiotherapy; additionally, anaerobic media were used. In the case of blood samples, routine tests were performed at various time points. Tests included morphology, liver tests, lipids, creatinine, urea, electrolytes, glucose, C-reactive protein (CRP), interleukin 6, coagulation tests, PSA.

### 2.4. Isolation and Culturing of Bacteria from Urine

Before microbiological examination, samples were defrosted at room temperature and then thoroughly vortexed. An amount of 0.5 mL of each urine sample was added to 4.5 mL sterile peptone water (Sigma Aldrich, Steinheim, Germany) to obtain the 10^−1^ dilution. Subsequently, 100 μL of undiluted and diluted urine samples was plated onto 3 different types of culture media: Tryptic Soy Agar (TSA; Sigma Aldrich, Steinheim, Germany), Schaedler Agar (SCH; Sigma Aldrich, Steinheim, Germany), and CLED Agar (CLED; Sigma Aldrich, Steinheim, Germany), followed by incubation at 37 °C for 18–24 h. All culture media were in the form of ready-to-use powders. For obtaining pure cultures, single colonies demonstrating different morphologies were transferred onto new plates, streaked, and incubated at 37 °C for 18–24 h.

### 2.5. Identification of Bacterial Colonies

For the isolates’ identification, matrix-assisted laser desorption ionization time of flight mass spectrometry (MALDI TOF MS technique), MALDI Biotyper 3.0 platform (Bruker Daltonics GmbH, Bremen, Germany), and bacterial protein extraction protocol according to Bruker’s guideline were used. Amounts of 300 µL of sterile deionized water and 1 inoculation loop (10 μL) of bacterial biomass were added to a 1.5 mL tube and mixed together using a vortex. Afterward, 900 μL of 96% ethyl alcohol was added, mixed again, and centrifuged (1300 rpm, 5 min). The supernatant was discarded, and the remaining cell pellet was dried using a vacuum centrifuge at room temperature. To extract proteins from bacterial cells, 10 µL of 70% formic acid and 10 µL acetonitrile were added to the cell pellet and mixed by pipetting. Then, the obtained extracts were centrifuged (1300 rpm, 5 min), and the supernatants were used for further analysis.

Before identification, 1 µL of supernatant was transferred onto a MALDI MTP 384 ground steel target sample spot (Bruker Daltonics GmbH, Bremen, Germany). After air drying, the samples were overlaid with 1 µL of MALDI matrix solution—10 mg/mL α-cyano-4-hydroxycinnamic acid (HCCA; Sigma Aldrich, Buchs, Switzerland) dissolved in standard solvent solution (50% acetonitrile, 47.5% water, and 2.5% trifluoroacetic acid).

Samples were analyzed using an ultrafleXtreme MALDI-TOF mass spectrometer (Bruker Daltonics GmbH, Bremen, Germany) equipped with the smartbeam-II laser–positive mode. Spectra were collected manually through manufacturer software flexControl, using parameters described in a previous work [14]. The collected spectra were subjected to smoothing, applying the Savitzky–Golay method, baseline corrections using the TopHat algorithm, and calibration with the use of Bruker’s bacterial test standard (Bruker Daltonik) in quadratic mode via the manufacturer software flexAnalysis. Each sample was measured at least in duplicate.

### 2.6. Statistical Analyses

Categorical tests were carried out in an R environment (RStudio v.1.1.463, Vienna, Austria). Chi-square test was performed using “chisq.test” R function to search for statistically significant differences between the expected and observed frequencies of bacteria isolated from urine samples obtained using different collection procedures and at different time points of the patient’s treatment course. In case the assessed expected frequencies were fewer than 5 for at least 20% of the variables, Fisher’s exact test was performed instead (“fisher.test” R function). Spearman correlation analysis (“corrplot” package) was conducted to observe possible relationships between the number of detected bacterial species, sampling time points, and biochemical parameters. The following analyses were conducted using IBM SPSS v.23 (Armonk, NY, USA). The Kolmogorov–Smirnov test was used to assess the normality of data distribution. Mann–Whitney (non-parametric) and t or ANOVA (parametric) tests were carried out, aiming to point out bacteria that presented significant differences in their incidences depending on the sampling time point and to assess alterations in biochemical parameters between the studied sample cohorts.

## 3. Results

### 3.1. Urinary Microbiome Characteristic

Microorganisms were present in 140/181 urine samples, using three different culture media obtained over 450 bacterial isolates. The best microbial growth was observed on TSA (universal medium) and CLED (differential medium for urine specimens); the largest number of isolates was obtained in those media. MALDI-TOF MS identification revealed the presence of 33 different bacterial species—3 Gram-negative (G−) and 30 Gram-positive (G+) (Figure 1).

The vast majority of the identified bacterial species were represented by the following families: *Staphylococcaceae* (39%), *Corynebacteriaceae* (15%), and *Streptococcaceae* (12%). Considering all the studied samples with bacteria, the most frequently isolated species were: *Staphylococcus haemolyticus* (in 60 samples out of 140; 42.9%), *Staphylococcus epidermidis* (56; 40.0%), *Staphylococcus hominis* (52; 37.1%), *Enterococcus faecalis* (42; 30.0%), and *Micrococcus luteus* (31; 22.1%). The following combinations of species were most often isolated in one sample: *S. epidermidis* + *S. haemolyticus* (six cases), *S. epidermidis + S. haemolyticus* + *S. hominis* (six), *E. faecalis* + *S. haemolyticus* + *S. hominis* (five), and *S. haemolyticus* + *S. hominis* (five). In total, 61% of all the isolated bacterial species were present in at least two patients. When analyzing the samples collected from each patient separately, the following bacteria were most often isolated: *S. epidermidis* (35 patients; 70%), *S. haemolyticus* (30; 60%), *S. hominis* (26; 52%), *M. luteus* (20; 40%), and *E. faecalis* (19; 38%). This means that they were isolated at least at one stage of the research from the same patient.

### 3.2. Urinary Microbiome in First and Middle Void

Urine samples were collected from 12 patients from the first-void (40 samples) and middle (39 samples) streams. When comparing the type of urine samples, bacteria were more often found in the samples from the first-void urine than from the midstream one. The absence of bacteria was verified in 12.5% of samples from the first-void urine and in 35.9% of samples from the midstream one. More detailed information is presented in Figure 2. On the other hand, bacterial associations comprising five species were evidenced only in the midstream samples. The frequencies of the detected bacterial species between the examined types of urine samples differed (the ratios of the number of detected species per number of samples corresponding to the first and middle stream were 1.80 and 1.71, consecutively); however, the indicated difference in the distribution of species’ occurrence was not significant (*p* = 0.85). When evaluating the frequencies of individual species in both urine streams, *Enterococcus faecalis* and *Corynebacterium accolens* were the bacteria that presented the most expressive differential distribution between urinary fractions, both presenting greater-than expected-actual counts in the first-void urine (in both cases, for the Mann–Whitney test, *p* = 0.047). The frequency of some species was related to the urine stream. *E. faecalis* and *S. hominis* were more frequent in the first-void urine than in the midstream one. On the other hand, *S. haemolyticus* was more common in midstream urine. The incidences of *S. epidermidis* and *M. luteus* in both samples were comparable.

### 3.3. Urinary Microbiome at the Variable Time Points

The prevalence of different bacteria in urine samples collected at different time points in the treatment course is shown in Table 2. One bacterial species per sample was prevalent among urine specimens collected before gold fiducial implantation into the prostate gland, as well as 1 month after radiotherapy. At time point 2 (before radiotherapy), two species of bacteria were most often isolated. At the end of radiotherapy, no bacteria were detected in over 35% of the analyzed samples. One month after radiotherapy, one bacterial species was most often isolated in the samples.

The differential distribution of species’ variety according to the investigated sampling time point is suggested; however, it cannot be considered statistically relevant (Fisher’s exact test, *p* = 0.19). The assessment of chi-square residuals (difference between the observed and expected values—Figure 3) evidenced a positive association between the presence of five species and time point 4 (1 month after radiotherapy), the presence of four species at time point 1 (before gold fiducial implantation), and the absence of species and time point 3 (at the end of RT). This trend reveals that at the beginning (time point 1) and later (time point 4) stages of the treatment course, the variety of bacteria present in the urinary tract was superior. At the intermediary sampling time points (before radiotherapy and at the end of radiotherapy), generally, a lower number of bacteria were detected in samples. Furthermore, Pearson’s correlation analysis pointed out that, for time points 2 (before radiotherapy) and 3 (at the end of RT), the number of species present in a bacteria association was negatively related to the counts of incident species (r = −0.88, *p* = 0.019, and r = −0.98, *p* = 2.2 × 10^−4^, respectively), demonstrating that the variety of bacterial species tends to be inversely proportional to the progression of the treatment. As a consequence, time points 2 and 3 appear strongly positively correlated (rho = 0.87, *p* = 0.023).

The total number of detected bacterial species was significantly higher at time point 1 in comparison to time point 3 (Mann–Whitney test, *p* = 0.038). Considering the incidence of individual bacterial genera at different time points, a greater alteration was indicated when comparing time points 1 and 3 for the Staphylococcus genus (Mann–Whitney test, *p* = 0.012), whose incidence was significantly superior at the first sampling time point.

### 3.4. Correlation between the Urinary Microbiome and Biochemical Parameters

One-way ANOVA was performed, aiming to identify variables with statistically relevant alterations in their means across samples collected at different time points. The related parameters as white blood cell count (WBC) (*p* = 0.009) and lymphocytes (*p* = 0.012) in blood showed to have their values altered depending on the evaluated sampling times. The greatest difference was observed between t1 and t4, and the numbers of WBC (*t*-test, *p* = 0.001) and lymphocytes (*t*-test, *p* = 0.011) were significantly higher at the first sampling time point. Analysis of Spearman’s rank correlation coefficient (Appendix A) revealed moderate to strong positive correlations between urine coloration range and urine specific gravity (rho = 0.37, *p* = 9.20 × 10^−5^), as well as the registered levels of leukocytes (rho = 0.328, *p* = 0.001) and proteins (rho = 0.627, *p* = 0.029). Urine specific gravity also showed to be indirectly linked to sample pH (rho = −0.423, *p* = 1.04 × 10^−5^). The number of bacterial species present in urine samples was also significantly correlated with sample coloration (rho = 0.231, *p* = 0.019, a weak but relevant relationship) and leukocytes concentration (rho = 0.366, *p* = 2.37 × 10^−4^, a moderate correlation). The progression of treatment with time showed to be negatively connected—at moderate strength—with the WBC parameter (rho = −0.441, *p* = 3.32 × 10^−4^) and other associated variables: the levels of neutrophils (rho = −0.303, *p* = 0.016), lymphocytes (rho = −0.368, *p* = 0.003), and monocytes (rho = −0.336, *p* = 0.007). Such observations may denote a decrease in inflammatory response as the radiotherapy treatment advances. In addition to that, sample-matched PSA concentrations presented a negative and strong correlation with red blood cell count (RBC) (rho = −0.797, *p* = 0.001) and hemoglobin amount (rho = −0.639, *p* = 0.018). Total radiotherapy dose was inversely linked to the following biochemical parameters: WBC (rho = −0.428, *p* = 0.018), neutrophils (rho = −0.367, *p* = 0.046), and C-reactive protein levels (rho = −0.433, *p* = 0.021).

Additional statistical examination was performed on PSA data recorded for the following time periods: before gold fiducial implantation into the prostate gland (t = 1), 1 month (t4), 4 months (t5) after radiotherapy. It was demonstrated that PSA levels displayed a strong negative correlation with the elapsed time of treatment (rho =−0.65, *p* = 2.13 × 10^−24^). Relative PSA levels (PSA at a referred time point/PSA at t1) were significantly lower in patients submitted to hormonotherapy for all the assessed treatment time points (for t4, t5 *p* = 4.84 × 10^−7^, and 7.61 × 10^−6^, respectively—according to the Mann–Whitney test). In addition to that, such relative PSA concentrations across time were statistically inferior in the patient group treated using the linear accelerator as a radiotherapy device compared to Cyberknife (for t4, t5 *p* = 2.28 × 10^−4^, and 0.031, consecutively). Within a short follow-up, this may be related to the frequent use of hormone therapy in the group of patients treated with the linear accelerator (only two patients were not on hormone therapy), in contrast to the patients treated with the Cyberknife, where 72% of patients were not on hormonal treatment.

## 4. Discussion

Identifying the individual microorganisms and the microbial community structure is an essential step in the diagnosis of various diseases. Recently, the link between cancer and disturbances in normal microbiota has become more clinically apparent. The association between cancer and the microbiome is not new [2]. The microbiome can promote tumorigenesis by generating chronic inflammation, disrupting the balance between cell death and proliferation, and triggering immune responses [15]. Infection agents cause up to 20% of cancers worldwide [2]. Still, the influence of the urinary microbiome on genitourinary cancer development has not yet been fully examined. Urine samples from healthy men or those diagnosed with prostate cancer showed no significant differences in diversity, but authors identified pro-inflammatory bacteria and observed some differences in uropathogens between cancer grades [9]. Additionally, Alanee et al. examined the association of the urinary and fecal microbiota with prostate cancer after transrectal prostate biopsy [10]. Analysis of the urine samples revealed a decreased abundance of several bacterial species in patients with prostate cancer.

Research indicates that the urine microbiome has not yet been clearly defined, but among healthy people, it consists mainly of slow-growing and fastidious microorganisms, mainly belonging to five phyla: *Firmicutes*, *Bacteroidetes*, *Actinobacteria*, *Fusobacteria*, and *Proteobacteria* [16]. Numerous studies show that there are differences between the female and male microbiome [6,17] and that it changes with age [18]. Among healthy men, the predominant genera of bacteria are *Corynebacterium*, *Staphylococcus*, and *Streptococcus* [17]. The results of the present study showed that the dominant bacteria in urine samples were members of the genera *Staphylococcus*, *Enterococcus,* and *Streptococcus.* Very similar results were obtained by Shrestha et al. who analyzed 135 urine samples from men after biopsies who were or were not diagnosed with prostate cancer [9]. *Corynebacterium*, *Staphylococcus*, and *Streptococcus* were the predominant bacteria in their research. The reason that there are differences in the microbiome may result from the different methods of microorganism identification. The published data only provide information about the use of the PCR technique in the analysis of urine among men with prostate cancer. The PCR technique, which was used in the study presented above, allows for the identification of microorganisms based on the DNA extracted from the urine. PCR methods are adequate for determining the total DNA in a sample, but they are not able to differentiate whether the DNA comes from the living or dead cells or extracellular DNA [19]. The use of the MALDI technique allows for the identification of live bacteria (because microorganisms have to be cultured in the first place). For this reason, its use in clinical trials seems more appropriate to search for the characteristic microbiome of various disease states, such as prostate cancer. This statement is supported by the latest work of Dubourg et al. (2020) who investigated the use of the MALDI TOF MS technique in the so-called culturomics approach for deciphering urinary microbiota [20]. The authors isolated 450 different bacterial species from 435 urine samples, among which 256 have never been described in urine, and 18 were new species. Moreover, the obtained results indicated the need to shift a paradigm about urinary microbiota, that is, many members of the microbiota in the urinary tract are derived from the gut. Coagulase-negative *Staphylococcus* strains (e.g., *S. haemolyticus*, *S. hominis, S. saprophyticus*, *S. epidermidis*) were the most common in our research. Most of them are opportunistic pathogens, since their presence in urine does not always indicate an infection, but they can be associated with hospital-acquired infections, e.g., after catheters and bacteremia [21]. *S. haemolyticus* is often part of the urine microbiome, especially among men [22]. Studies imply that the urinary tract harbors a unique urinary microbiota, which is substantially different from the populations of the reproductive system and the gut [23]. It is also known that patients with urinary tract infection should demonstrate not only bacteriuria but also pyuria. Pyuria, which indicates an inflammatory reaction in the urinary tract, is generally defined as a positive leukocyte esterase on urine dipstick or ≥10 white blood cells per high-powered field (WBCs/hpf) by urine microscopy [24,25]. Our results of the correlation between blood and urine biochemical parameters with the microbiological outcomes revealed that the presence of leukocytes in urine was accompanied by higher bacterial species variety.

However, urine samples can suffer from skin and perineal contamination if the periurethral area is not sterilized before urine collection [16]. The prostate, even among healthy men, is inhabited by a small number of Gram-positive bacterial species, such as *S. epidermidis* [26]. Among isolates that were collected in this study, there were bacteria considered to be uropathogens. *S. saprophyticus* infections mostly cause UTI in young women, but in males, they have been associated with urethritis and prostatitis [27]. Three of the isolated bacteria (*S. hominis*, *M. luteus*, and *S. warneri*) are considered to be potential probiotic strains. They produce peptides and enzymes, which give them antibacterial activities against pathogens, such as *Escherichia coli* or *Staphylococcus aureus* [28]. In our study, no growth of these pathogens (*E. coli* and *S. aureus*) was observed in any of the samples with at least one of the probiotic strains mentioned above (over 46% of samples).

*E. faecalis,* which was isolated in over one-third of the patients, is considered the second most important cause of urinary tract infections (UTI). This bacterium is part of the fecal microflora and can cause UTI through an endogenous route [29]. This bacterium was more frequently present in the first-void urine than in the middle stream, suggesting that the samples could be contaminated. In the case of the remaining bacterial strains, no significant differences in their occurrence in the first and middle urine streams were observed. However, *Streptococcus* and *Staphylococcus* species were more frequently isolated in middle urine. These types are indicated in other studies as most often isolated from urine samples of healthy men [17]. This observation seems to be beneficial to patients, considering that they are not always able to collect urine properly, despite the instructions. Since there are no large discrepancies, the collected stream does not significantly affect the identification result of the bladder microbiome. This information is extremely important for future research into the correlation of changes in the urinary microbiome with radiation side effects.

There is a change in the number of bacteria species concerning the time of the sample collection. At the intermediary sampling time points (before radiotherapy and at the end of radiotherapy), generally, lower numbers of bacteria were detected in samples, indicating that the procedures before radiotherapy and radiotherapy itself result in depleting the local microbiome. At time point 1, before fiducial implantation, the samples represent the general urinary microbiome status in patients. Fiducial implantation requires the use of antibiotics that induce a decrease in the number/species of bacteria on the day of radiotherapy beginning. Antibiotics are the most popular treatment for urinary tract infections and are also used as a prophylactic treatment. However, the use of broad-spectrum antibiotics can negatively influence the beneficial microflora in the patient and consequential selective overgrowth of pathogenic bacteria. The exact effects of antibiotics on urinary microbiota are not known, but clinical trials are ongoing on this topic. Additional treatment with radiation also induces depletion of bacterial species if there is no co-infection associated with the occurrence of radiation side effects, such as difficulty in urinating and urine retention in the bladder. During the research, it was observed that the applied treatments, both RT and the hormone therapy, contributed to the overall decline in the diversity of the microbiome. Due to the small number of individual isolates, it is not possible to specify the changes in urine microbes resulting from the addition of androgen deprivation therapy (hormonotherapy). Including more patients in subsequent studies would make it much easier to find changes in the occurrence of individual strains in urine samples. In our group of patients, the most frequently isolated strains were *S. haemolyticus, S. epidermidis, S. hominis, E. faecalis, and M. luteus*. It can be noticed that, only in the case of *E. faecalis*, no significant changes in the frequency of its occurrence were observed, which may indicate accidental contamination of samples by patients during their collection. In the case of the remaining strains, their presence in the urine at the end of RT was approximately 15% lower. The decreased number/species of bacteria was restored 1 month after irradiation. This suggests that our radiation treatment does not completely damage the urinary microflora, allowing regeneration. Four months after RT, *S. hominis* and *S. haemolyticus* were detected in approximately half of the collected samples. The remaining strains were found in a much smaller percentage. Moreover, studies show that differences in the number and diversity of bacteria are more important than individual types found in patients. This is because the composition of the microbiome is influenced by many different factors, such as genetic predisposition, environment, diet, and the use of antibiotics.

Microbiome is extensively analyzed in the gut and its changes in variable situations, but little information is available on the microbiome and urinary tract in cancer patients. To our knowledge, this is the first study evaluating the differences in bacterial species in men with prostate cancer undergoing radiotherapy. Unfortunately, this study has its limitations due to an insufficient number of patients required to assess microbiome differences, taking into account different doses of radiotherapy or hormonal therapy usage, local advancement of the disease, and the Gleason score. We are aware that all of these factors influence the oncological response and, at the same time, can affect the microbiome. Here, we present the first result of an ongoing project in which we intend to include 300 patients. A larger number of patients will allow us to assess the differences between the variable groups of patients. For now, the obtained results emphasize the need for further research on changes in the microbiome in urine, both during and after oncological treatment, in order to correlate its changes with the early and late side effects of radiotherapy and the clinical outcome of the disease.

## 5. Conclusions

Application of the MALDI identification appeared to be a suitable tool for the fast deciphering of differences in the microbial diversity of the urinary microbiota depending on the time point of radiotherapy, as well as between the type of urine specimens used—first-void and middle stream. The obtained results revealed a similar spectrum of bacteria from the initial and middle urine streams. In turn, the treatment of prostate cancer patients with both antibiotics before gold fiducial implantation and radiotherapy affected the bacterial species composition, namely depleting the urine microbiome, which recovered after ending the therapy. To our knowledge, this is the first study evaluating the differential bacterial species in men with prostate cancer undergoing radiotherapy. Further research should focus on the correlation of microbiome changes and their potential impact on radiation side effects in prostate cancer patients.

## Figures and Tables

**Figure 1 biomedicines-10-01806-f001:**
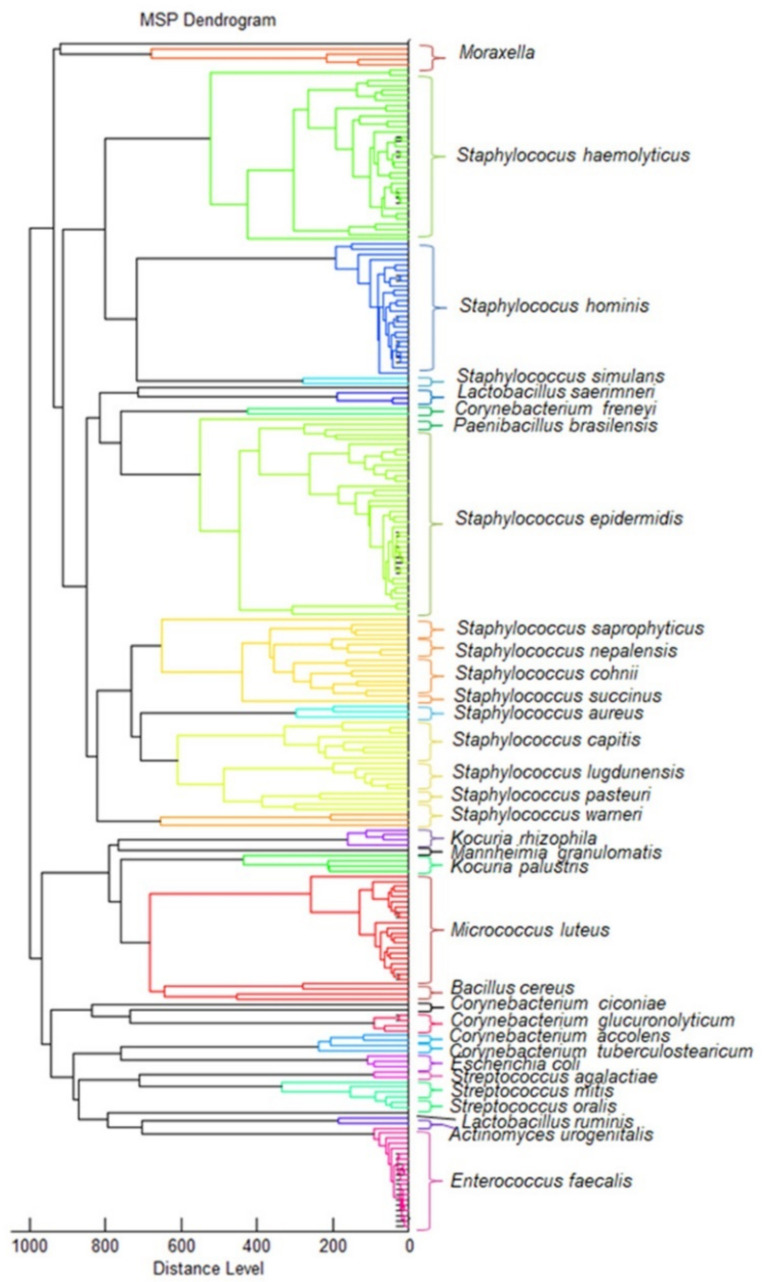
Main-spectra-profiles-based (MSP) dendrogram of all the bacteria isolated in this project. The dendrogram shows the relationship of the isolated strains on the basis of the proteome composition. The distance level reflects the degree of similarity—the smaller distance isolates are more closely related to each other. Strains of the same species are marked in one color.

**Figure 2 biomedicines-10-01806-f002:**
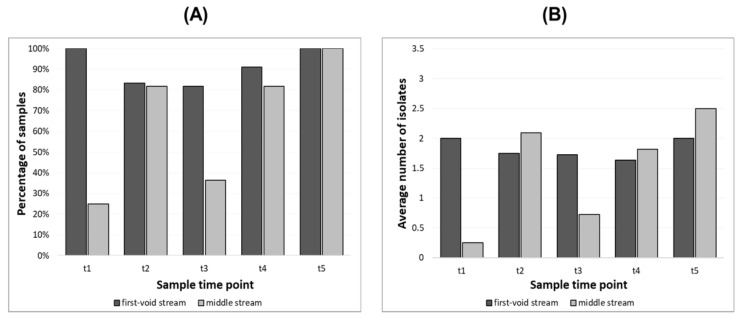
(**A**) The percentage of samples from which bacteria were isolated at the variable time points. (**B**) The average number of isolates isolated from individual urine samples at the variable time points. Variable time points: t1—before gold fiducial implantation into the prostate gland, t2—before radiotherapy, t3—at the end of radiotherapy, t4—1 month after radiotherapy, t5—4 months after radiotherapy.

**Figure 3 biomedicines-10-01806-f003:**
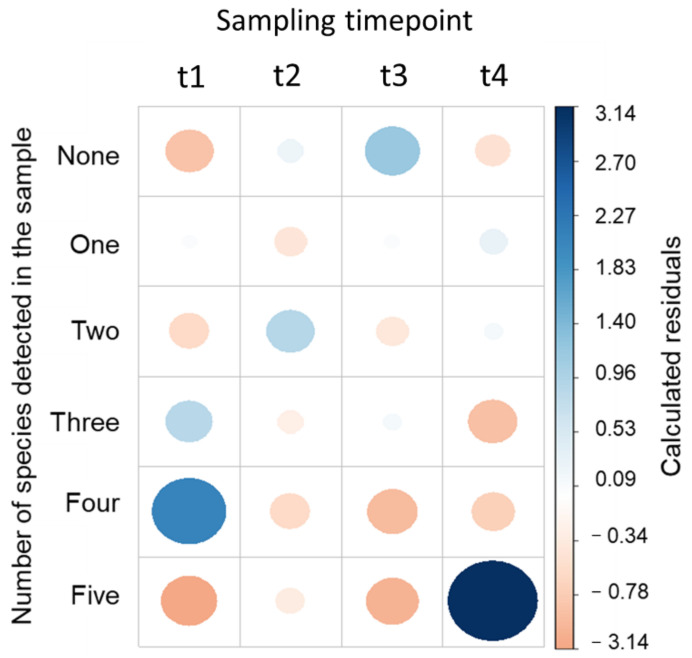
Plot of chi-square residuals (correlation coefficient) for the bacterial associations found at different sampling time points (t1—before gold fiducial implantation; t2—before radiotherapy; t3—at the end of the RT; t4—1 month after radiotherapy). The color and circle size refer to the degree of correlation coefficient. Blue color refers to the positive correlation, while red color refers to negative correlation.

**Table 1 biomedicines-10-01806-t001:** Patient characteristics. ^a^ In two patients, we did not obtain information about the baseline PSA levels before hormonal therapy was introduced by the urologist.

Patient Characteristics	N/%
Median age, 68 (range 51–83)	
Clinical stage	
T1a–T1c	26 (52%)
T2a–T2c	20 (40%)
T3–T4	4 (8%)
N0	44 (88%)
PSA	
<10	27 (54%)
10–20	13 (26%)
>20	8 (16%)
Not known ^a^	2 (4%)
PSA before radiotherapy median (range)	9.16 (4.1–87.3)
PSA 1 month after radiotherapy median (range)	0.204 (0.004–8.35)
PSA 4 months after radiotherapy median (range)	0.049 (0.004–6.82)
Gleason score	
6	24 (48%)
7 (3 + 4)	12 (24%)
7 (4 + 3)	3 (6%)
8	9 (18%)
9–10	2 (4%)
Use of androgen deprivation therapy	
Yes	30 (60%)
No	20 (40%)
Treatment/dose	
Linear accelerator 50–78/2 Gy fx	25 (50%)
Cyberknife 36.25/7.25 Gy fx	25 (50%)
Irradiated volume	
Prostate alone	28 (56%)
Prostate bed and pelvic lymph nodes	2 (4%)
Prostate and pelvic lymph nodes	20 (40%)

**Table 2 biomedicines-10-01806-t002:** Frequency of bacterial associations (number of bacterial species simultaneously detected in a sample) found in samples collected at different time points of treatment course.

Number of Bacterial Species	Frequency in Time Points [%]			
	Before Marker Implantation	Before Radiotherapy	At the End of Radiotherapy	1 Month after Radiotherapy
none	18.0	27.0	35.5	20.0
one	25.6	21.6	25.8	28.0
two	18.0	29.7	19.4	24.0
three	20.5	13.5	16.1	8.0
four	18.0	5.4	3.2	4.0
five	0.0	2.7	0.0	16.0

## Data Availability

All data generated or analyzed in this study are included in this published article (and its Appendix A).

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
