# Peer review of "A New Approach to Imaging and Rapid Microbiome Identification for Prostate Cancer Patients Undergoing Radiotherapy"

_biomedicines, 2022, doi:10.3390/biomedicines10081806_

Round 1

Reviewer 1 Report

Novel and interesting study.

The introduction is quite clear, not so clear the aim of the study. I believe that the main endpoint should be evaluate if there is any change in the microbiota composition during the different phases of the RT treatment. In the intro it seems that actually the main endpoint is only to evaluate the microbiota before RT treatment which is not so interesting, and the secondary endpoint is the evaluation of possible microbiota change during and after the treatment.

Would have been interesting to know which species are those who showed the huge increase/decrease during the treatment

Limitations of the study should be highlighted and declared ad the end of the discussion. For example:

-        Limited sample size

-        some patients took hormonal therapy and other not. The ADT might influence the microbiota.

-        Two different types of RT: line accelerator and cyber knife

-        Different Gy doses

-        etc

Reviewer 2 Report

The article is devoted to an urgent topic, the search for potential predictors associated with the effectiveness of radiotherapy. However, the discussion and conclusion to the article requires careful processing. It is necessary to clearly identify the identified changes taking into account changes in biochemical markers and PSA content.

Round 2

Reviewer 1 Report

I read the new version and the authors replied to my concerns.